# Factors That Determine Successful Social Housing of African Green Monkeys (*Chlorocebus sabaeus*) in Same-Sex Pairs and Trios

**DOI:** 10.3390/vetsci11120667

**Published:** 2024-12-20

**Authors:** Amanda M. Murti, Clive C. Wilson, Antonio F. Pemberton, Tatiana M. Corey, Loveness N. Dzikiti, John D. Elsworth, Calvin B. Carpenter

**Affiliations:** 1Virscio, Inc. 4 Science Park, New Haven, CT 06511, USA; 2St. Kitts Biomedical Research Foundation, Lower Bourryeau Estate, Basseterre 00265, Saint Kitts and Nevis; 3University of Texas Health Science Center at San Antonio, 7703 Floyd Curl Dr, San Antonio, TX 78229, USA

**Keywords:** African green monkey, St. Kitts, pair, trio, social housing, environment, enrichment, welfare, refinement

## Abstract

African green monkeys (*Chlorocebus sabaeus)* have gained interest as an alternative non-human primate research species to the more commonly used macaque species. This is due to the current non-human primate shortage for valuable participants in critical biomedical research. African greens are a social species that require social housing for their health and welfare. We developed a method of social partner(s) selection evaluating sex, age, weight, cage size, and configuration to create successful pairs and trios for animals participating in research studies. This method is tailored for a high-throughput contract research organization where thorough temperament testing is not always possible. Both females and males showed high rates of success in pair housing; however, females showed greater success in trio housing compared to males. We hope these methods will help other research institutions create lasting successful social groups of African green monkeys.

## 1. Introduction

The current shortage of research non-human primates (NHPs) has elevated interest in African green monkeys (*Chlorocebus sabaeus*) [1]. On the Caribbean islands of St. Kitts and Nevis, there are an estimated 37,000 free-roaming African green monkeys based on the most recent census [2]. A small founding population of African green monkeys was introduced to the islands in the 17th century, and the population has greatly expanded, such that they have become an invasive species, creating major problems for the local agricultural industry [3]. Humane procurement to enlist for biomedical research contributes to population management and allows for research husbandry to occur within a primate source country where animals can be housed in an outdoor naturalistic setting. Consequently, we have sought to optimize strategies for establishing safe and effective social housing for African green monkeys that are housed in research facilities.

The freedom to participate in social interactions is one of the most important enrichments available for NHPs housed in captivity [4,5]. The African green monkey is a social species, and isolation can quickly cause an increase in stress and lead to immunocompromise [6,7]. This is both a serious welfare issue as well as a factor that can compromise study data [8,9,10]. Compared with single housing, social housing can lead to a better appetite, faster recovery from study procedures, and improved overall health as reflected by data from bloodwork and physical exams [11,12,13,14]. However, there is currently limited published information about procedures for initiating and maintaining pair housing for African green monkeys and no information about trio housing. As their contributions as a research species are increasing, it is important to document the behavioral differences between Caribbean-origin African green monkeys and, presently, the more commonly enlisted macaque species. Comparatively, green monkeys originating from St. Kitts and Nevis are more reserved, behaving more like a prey species hiding vulnerabilities [15,16,17], often showing more subtleties in behavior and less rigid dominance hierarchies compared to macaques [18,19,20]. Another distinctive behavior of African greens is territoriality and fierce defense of their home ranges [15,21], a behavior that is also demonstrated in the setting of home enclosures in captivity [16]. Unlike macaques, African greens are proportionately more arboreal than terrestrial, making vertical space an important component of not just their environmental enrichment but critical to social dynamics and overall behavioral health [16,22,23,24]. While macaques benefit from a gradual introduction to new social partners through protective barriers, this is not true for African greens. It has been theorized that the species requires full physical contact to enable normal behavioral interactions surrounding a social introduction [16,25,26].

Many studies are designed to have treatment groups containing three animals each, which previously has made pair housing a challenge, as only two of the three could be paired in a group. Same-sex social groups are preferred for most pharmacokinetic, pharmacodynamic, or behavioral studies conducted, due to the variable impact that sexual interactions could have on maintaining stable biochemistry and behavior. Consequently, our emphasis has been on creating compatible same-sex treatment groups of African green monkeys that are socially housed for studies that may last for a year or longer.

The current study aimed to delineate which factors during the introductory period of the social group would optimize successful establishment of male or female trios. A second objective was to quantify the success rate of pair and trio social groupings of either female or males using these procedures. Our hypothesis was that for both female and male groupings, those with a greater body weight differential within the group would be more successful than those with equivalent weights, as one animal would likely be rapidly identified as the dominant animal. It was also hypothesized that groups provided with a larger cage size would be more successful, as it provided more space to perform introductory behaviors. Finally, based on the historical observations of pair housing in our colony, it was hypothesized that the female trios would be more successful than the males.

## 2. Materials and Methods

### 2.1. Facility Organization of Animal Housing

The animals are housed at a contract research organization with NHP facilities located on the island of St. Kitts, operating at the Saint Kitts Biomedical Research Foundation. The facility is situated on former agricultural land with substantial research infrastructure that currently accommodates up to 1500 African green monkeys, which includes a combination of wild-caught and purpose-bred animals. There are typically about 700 animals housed in large outdoor social enclosures (Figure 1).

There are multiple types of animal groups housed in these social enclosures: all adult males or females (about 10–12 animals/enclosure), all juvenile males or females (about 12–14 animals/enclosure), and breeding enclosures (1 male with about 6–7 females, along with their juvenile and infant offspring). The facility also has enclosures specifically designated for same-sex geriatric animals or for dams and young infants that have not yet been assigned to an established breeding group.

Within the testing facility there are typically about 500 animals participating in studies at any one time. These animals are housed in study buildings, which are built using plastic lumber with slatted walls to promote natural breeze ventilation and visual and auditory connections to each other and the outside environment (Figure 2). Further information on the facility, intake, and quarantine procedures are elaborated on in Appendix A.

When animals are enrolled in a study, all will participate in the social housing program, unless there is an IACUC-approved or veterinary reason for single housing. This often involves grouping animals that have been assigned to the same treatment group of a study. Frequently, these groups contain two or three animals, so there are very limited options for selecting partners among those screened for participation in the study. There are a wide array of study types conducted, including ophthalmic, neurological, cardiovascular, metabolic, and toxicological studies, and the study type and design will determine how the treatment groups are established. For example, many gene therapy studies involve use of viral vectors, where theoretically, a virus could be spread to a cage partner, and social housing should not mix different treatment groups.

### 2.2. Establishment of Animal Groupings

Our social housing method aims to combine similarly aged males or females that have a clear weight discrepancy, ideally at least a 0.5–1 kg difference, in an enclosure that is novel to these animals to avoid territorial aggression. A novel cage setup in this context is referring to two or more connected individual cages (Figure 2, Figure 3 and Figure 4) that have not been previously lived in by any of the members of the new pair or trio. For adult males, introduction is initiated after sedation with intramuscular ketamine alone (5–8 mg/kg) or ketamine and xylazine (8 mg/kg and 1.6 mg/kg, respectively). Ketamine alone is the preferred sedation method for introductions. However, a ketamine/xylazine cocktail is the common sedation method for performing study procedures. To avoid multiple sedation events unnecessarily, pairs or trios may be introduced opportunistically while sedated with a ketamine/xylazine cocktail for other procedures. Animals are observed by veterinary staff as they recover in this novel enclosure. The social housing method is detailed further in Appendix B.

The success of pairs or trios is determined based on a compatibility assessment of a group starting during the introductory period and continuing throughout the course of the study (Table 1). If signs of incompatibility are noted, a group is evaluated to determine if separation is warranted (Table 2). A pair or trio that needs to be permanently separated during a study is determined to be unsuccessful. Permanent separation is determined based on the frequency and level of aggressive behaviors noted in a group and any emotional distress from one or more of the group members (Table 2). In some cases, though, a group may be separated early due to concerns from the study team about occasional contact aggression (Table 2, Appendix C) that develops during the introduction period. Pair/trio housing is not terminated simply due to contact aggression, however. Decisions of intervention must balance the welfare of the animal with the monkeys’ need to establish social relationships and hierarchies between themselves. Previous studies at other institutions have used 14 days after the initiation of full contact as a time point that signifies success of a social group [27]; however, in this dataset, we wanted to define success as maintaining a social group throughout the entirety of a study, as there often are no other social housing options if the treatment group in question fails as a pair/trio. Studies can range from 3 months to 1 year long, and animals in non-terminal studies can remain in this successful social group after the conclusion of the study.

### 2.3. Statistical Analysis of Social Housing Methods

#### 2.3.1. Retrospective Analysis—Historical Method

A retrospective analysis of pairing data from 1986 to 2023 was conducted based on pairing statuses entered in the medical records database (DVMAX) by veterinary staff. Four medical record codes were used in the analysis: pair housing—no mesh divider/full contact, pair housing—mesh divider/protected contact, pair housing—separated, pair housing—removed due to aggression. “Pair housing—separated” includes multiple reasons for incompatibility (Table 2), and “pair housing—removed due to aggression” will specify which of the pair members was instigating the aggression. For this analysis, both “pair housing—separated” and “pair housing—removed due to aggression” were grouped together to be considered a failed pairing attempt. Data were analyzed from the first pairing attempt of 443 animals. Details on the participating animals are included in Table 3. A successful pairing in this dataset was defined as a pair that did not need to be permanently separated. If during the introductory period or any time afterwards, a pair needed to be temporarily separated but was able to be successfully reintroduced, that was also considered as a success. Successful pairing was separated into “full contact” and “protected contact”. “Full contact” is the ideal scenario, where there are no barriers dividing the two animals. “Protected contact” provides a mesh barrier between the two animals, which still allows for visual and tactile contact through the divider. Animals labeled as “protected contact” in the medical record were able to coexist next to each other, showing no signs of distress once the mesh divider was in place (Table 2). An animal that showed any signs of distress during “protected contact” was moved and recorded as “separated”. Protected contact was most often employed in animals that were not able to be given full contact due to research study constraints but allowed the animals to have some tactile social enrichment during that study. Protected contact was also used on a case-by-case basis if a pair of animals was displaying some signs of incompatibility when given full access but were showing no signs of distress as neighbors, which allowed for a higher degree of social enrichment than full separation.

#### 2.3.2. New Method for Male Pair Housing

Between 2022 and 2024, 20 male pairs were created on research studies (40 total male monkeys). All pairs were given additional cage space, with two top cages and one lower cage, or two bottom cages and one top cage, in an L configuration (Figure 3). In some instances, the males were able to be given all four cages in the quad, at least during the first two weeks of the introductory period (Figure 4). After the introductory period, if building constraints required, the pair was brought back to having two connected horizontal cages (no earlier than four weeks after introduction), either the top two or bottom two cages (Figure 2).

#### 2.3.3. Trio Housing Method

Starting in 2022, the established method of social housing was modified to include trio housing, as many study treatment groups include three same-sex members that could not be housed with members of another treatment group. To achieve trio housing, different modifications of the pair housing procedure were instituted for each sex.

For the analysis reported here, all female and male trios that were created for research studies from 2022 to 2024 were analyzed for group success, as defined above, throughout the course of that study. The number of male and female trios were necessarily constrained by the designs of the studies conducted during that period. Due to these constraints, there were an uneven number of female and male trios analyzed. More female trios (*n* = 25) were created during this period (75 total female monkeys) than male trios (*n* = 11) (33 total male monkeys).

For females, most trios initially underwent sedation with intramuscular ketamine alone (5–8 mg/kg) or ketamine and xylazine (8 mg/kg and 1.6 mg/kg, respectively) while moved to novel cages in the same study building. For animals that were given the ketamine/xylazine cocktail instead of ketamine alone, this was because introductions were performed on the same day as baseline activities, as this already scheduled sedation event aided in animal movement and cage set up. Thus, to avoid an additional sedation event for social grouping, the groups were often opportunistically created on this day. For the female trios in Study 1 (Table 4), who were the first four female trios created, groups were introduced without sedation because of historical evidence of success for awake female pairing introductions. Space constraints in the study building did not allow for novel caging to be provided for these females, so they were grouped together by pulling the dividers between the females in their home cages while awake. These females had existed in protected contact for at least one week without any signs of incompatibility (Table 2) before being moved to full contact. All subsequent female trios in this dataset were sedated and moved to novel caging.

If there was sufficient cage space available, each female trio group was given an additional cage, so that the three animals were housed in four cages. In these instances, the quad of four cages (two top cages and two bottom cages) was organized in a C-formation (Figure 4). The floor of one of the top cages was removed; this cage still had perches present and often was a favorite spot for the animals. For the female trios, the variables of sedation, novel caging, caging configuration, age difference, age group, and weight difference were all recorded, but the variables could not be meaningfully analyzed for their impact on the outcome since all trios were successes in this dataset.

All male trios in our study were either sedated with intramuscular ketamine alone (5–8 mg/kg) or ketamine and xylazine (8 mg/kg and 1.6 mg/kg, respectively) to initiate social housing. All male trios were introduced in caging that was novel/neutral for each member. Whenever possible, males were housed in (*n* + 1) caging, organized in the C formation (Figure 4). In two of the male trios in Study 15 (Table 4), this was not possible due to constraints in the room, so in these cases, three animals occupied three cages in an upside-down L formation (Figure 3). In one other male trio in Study 12 (Table 4), the C-formation was not possible due to constraints in the room, so the three males were given four cages connected horizontally, with sight barriers between each one. For the male trios, the variables of caging configuration, age difference, age group and weight difference were all analyzed.

#### 2.3.4. Statistical Analysis Statement

To assess the effectiveness of the strategies for establishing pairs and trios, percent success was calculated based on sex and cage space. For males paired using the new method, *t*-tests were performed to analyze whether weight difference and age difference influenced success of the pair. The weight difference and age difference were analyzed as independent variables. A Fisher’s exact test was performed to analyze whether the age group of the pair members influenced success. The age groups were defined as juvenile (estimated 0–4 years old), adult (estimated 5–12 years old), or mixed juvenile and adult. For male trios, *t*-tests were performed to assess whether weight difference and age difference influenced trio success. The Fisher’s exact test was performed to analyze whether the cage type (*n* + 1 caging) or the age group of the trio members influenced success. *p* values of less than 0.05 were considered statistically significant.

### 2.4. Study Design Limitations

The pairs and trios created for this study were participating in research studies with existing IACUC-approved protocols dictating the length of the study, schedule of study activities, sexes, ages, and body weights, as well as the structure of the treatment groups. As such, there were some aspects of the study design that could not be controlled, as the social groups were created as part of these ongoing research studies. Within the weight ranges the studies allowed, groups were created with a weight discrepancy whenever possible. Determining a social group failure while enrolled on a research study was a limitation, as some groups may have been successful with more time, more space, or less study activity, but the social groups needed to fit into the ongoing study designs and timelines.

## 3. Results

### 3.1. Retrospective Analysis of Historical Method

Overall, for both males and females, most animals participating in the pair-housing program were successfully paired, with over 80% success rate with full contact (Table 3 and Figure 5 and Figure 6). For females, the overall success rate was better than for males, with over 90% success with full contact (Table 3 and Figure 5 and Figure 6). The success rate of full contact male pairs was lower than with females but still had almost 70% success (Table 3 and Figure 5 and Figure 6).

For both males and females, only a small percentage of animals needed to be removed completely from their pairs (Table 3). Some pairs that were not as compatible with full contact, and for which no other pairing options were available at the time, were able to have protected contact with their pair and live compatibly as neighbors. Protected contact still allowed for these animals to have some physical contact with their neighbor, to allow for the benefits of enriched social interaction, but also to provide them protection from any contact aggression that might occur. These animals were also considered successful pairs, as they still had social interaction, but were included separately in the percent success (Figure 6).

### 3.2. Male Pair Housing: New Method

In total, 18 out of 20 male pairs using the new method were successful (Figure 7), giving an overall success rate of 90% (Table 4). Age difference and weight difference appeared not to influence pairing success for males, with *p* = 0.97 and *p* = 0.16, respectively. Age group did not show a statistically significant association with pairing success, *p* = 1. The sample sizes in our study were small; larger sample sizes may show some statistical significance.

### 3.3. Trio Housing

Socially housed female trios had a 100% success rate, whereas male trio success was substantially lower than for female trios or male pairs (Table 4, Figure 8 and Figure 9). It should be noted that for male trios, two used the old method of one animal per cage (Study 15), and for trio housing, this was unsuccessful in both cases, as it resulted in contact aggression. All animal injuries were assessed by the veterinarians as minor wounds. However, study personnel were concerned about leaving the groups together for a longer period of observation while on the study in case aggressive behaviors escalated. In these instances, in Study 15, the member of the trio observed by the behavioral staff and veterinary staff to be instigating the aggressive behaviors was shifted to protected contact where they were able to interact with the remaining pair safely, without any observed signs of distress (Table 2).

The other male trios that used the new *n* + 1 (or *n* + 3) caging method were mostly successful (66.6%). None of the trios in our study ever inflicted serious wounds on any of the other animals (Table 4). For male trios, *t*-tests were performed to determine the influence of weight difference and age difference on trio success, and Fisher’s exact tests were performed to determine the influence of cage type and age group on success. Weight difference (*p* = 0.42) and cage type (*p* = 0.24) were not statistically significant in this dataset; however, with a larger sample size, these variables may be found to influence success. It appears that age difference is a significant contributor to male trio success (*p* = 0.02), in that the larger the difference in age, the less likely that a male trio will be successful. The males that were closer in age were the most successful. It also appears that age group is a significant contributor to male trio success (*p* = 0.015), in that the male trios were more likely to be successful in the juvenile age group. For this dataset, the juvenile males who were close in age were the most successful.

For female trios, all the trio housing attempts were successful, so we were unable to determine statistically whether success was influenced by age difference, weight difference, age group, or cage type; however, for this dataset of female trios, the majority were close in age, with only two trios having greater than a 2-year difference in age. The majority were in the adult age group, except for five, who were juvenile groups, and the majority were close in weight as well, with all having less than 1 kg difference except for two groups. The other variables to consider were that four female trios were introduced without sedation and were not able to be given novel caging. As all trios were equally successful, it appears that these variables may not influence trio-housing success in females.

## 4. Discussion

### 4.1. Historical Pairing Method

This study evaluated three different methods of social housing for the African green monkey at animal facilities on the Caribbean island of St. Kitts. The first method was our facility’s historical pair-housing method, which involved creating same-sex pairs with a significant weight difference between members (at least 0.5–1 kg) and sedating the animals before introducing them to one another in a novel caging environment. Overall, this method has been very successful in creating lasting pairs of females (90% success). For male pairs, while still successful (69% success), this study sought to evaluate a modification of the historical approach to improve success. Based on the territorial nature of the African green and the importance of vertical space for this semi-arboreal species [16,21,22,23,28], a new method for male pairing was devised.

### 4.2. New Pairing Method

This new method hypothesized that by providing at least one additional cage for these pairs during the introductory phase (2–4 weeks) and including vertical space (Figure 3 and Figure 4), the pairing success could be improved by creating conditions more appropriate for the natural behaviors needed for a social introduction. Other factors that were evaluated were the weight difference between members, the age difference between members, and the age group of the members (juvenile, adult, or mixed). For testing this new method, 20 male pairs on a variety of different research studies were evaluated (40 total animals), with a total percent success of 90%, which was a great improvement. Due to the nature of the research studies being conducted, the male animals participating in the studies had been selected within a narrow range for weight and age, to control these variables in these studies. Likely due to these constraints and the smaller sample size, the weight difference, age difference, and age group did not statistically influence male pair success in this dataset. Additional studies need to perform a more controlled comparison to evaluate the degree to which these factors influence male pair success. However, the high success rate using this new method within a narrower weight and age range is encouraging for other research organizations providing social enrichment for male African greens participating in similar research studies.

Providing the larger cage space with an ample vertical component, even just temporarily during the introductory period, resulted in a significant increase in pair success compared to the historical method, allowing for more animals to receive all the benefits of social enrichment. The increase in cage space in these successful pairs was able to be reduced back to two horizontal cages per pair after the introductory period had passed in some of the pairs, which, in our study, was at least four weeks after successful introduction. This reduction in space, which was due to building needs, was performed in conjunction with the animal care behavioral staff, research staff, and veterinary staff, with frequent observation for any adverse responses. McGuire [29] described wild-caught African greens from St. Kitts being relocated from a larger enclosure in St. Kitts to a smaller enclosure in California and noted that the smaller space increased affiliative behaviors and had no effect on aggression. To our knowledge, this is the only known study in African greens specifically looking at cage space and the effect on group social interactions; however, this involved much larger groups being moved to a smaller novel environment. We observed a significant increase in success with male pairs that were provided larger cage space in a novel cage environment. We theorize that this is especially critical during the introductory phase, but once the group is established and successful, it appears that reducing the space for these successful pairs does not contribute to any aggressive behaviors.

Further research is needed to determine the earliest point at which space reduction could be performed, if necessary, as this was not able to be controlled for in our study. Other factors that were not able to be controlled for in our study but warrant further investigation for their influences on pairing success are frequency of temporary separations of the pairs for study activities and frequency of sedation events needed for study activities, as it is theorized that interrupting the introductory period with these episodes may lead to pair failures.

### 4.3. Trio Housing

Trio housing was established at Virscio in 2022. This was a beneficial enhancement to the animal care program, as many research projects required separate treatment groups of three same-sex animals, and this housing improvement ensured the facility had the ability to house each group of three socially. In our study, all trio-housed males and females enrolled in research studies were analyzed for the success of group housing based on age difference, age group, weight difference, and the amount of space provided. Other factors include the configuration of the caging provided and for some females, whether the group members were sedated for introduction or moved to novel caging. It was observed that regardless of awake or sedated introductions, novel caging, age difference, or weight difference, all female trios were successful for the duration of the studies they were enrolled in. As is the nature of many of the research studies conducted at contract research organizations, there was a narrow range of ages and weights for the animals making up each trio. We were unable to interpret the statistical influence on success for these trios, as all were successful. Further research is needed to fully investigate the effect of these variables on successful social housing.

For the male trios assessed in our study, age group and age difference were determined to be statistically significant contributors to success. The juvenile and young adult males that were close in age were the most successful in trio housing, either with *n* + 1 caging or *n* + 3 caging.

Mature adult males could be difficult to maintain in trio housing for an extended period. The unsuccessful male trios were often able to maintain compatibility for about two weeks of trio housing, but as these animals were all on studies, it is theorized that the frequent separations for study activities may have interfered with the solidification of social bonds. Further research is needed to investigate to what degree frequent separation can affect these social groups.

For young adult and juvenile males, when all in a similar age range, adding at least one additional cage increased their success (e.g., three males with four cages). The most successful male trios lasted together for six months before study termination and included males about 4 years old, with caging that allowed each trio to have three additional cages (i.e., three males with six cages). These young males in this 6-month study were also too young for canine recontouring, so these males still had their primary canines with mixed dentition, while all adult males on studies had recontoured canines. Canine length has been attributed to higher social dominance rank in multiple NHPs [30,31], but it is unclear if canine length may have influenced the success of these juvenile trios.

It has been theorized that for African greens, a period of protected contact with visual access could cause frustration, in both males and females, and hinder the normal introductory process that involves physical contact [27]. Historically, we have observed similar findings specifically with adult males. Overall, based on the historical findings, none of the adult males that had any prior protected contact were chosen to be grouped together, to give the groups the best chance of success. However, after full contact introductions, in many cases, unsuccessful males were able to live with a mesh barrier between them, with no signs of aggression or distress. While there is no guarantee that two males will be able to coexist compatibly in protected contact after the introductory period, many males have been able to be housed as neighbors with no signs of distress. As this protected contact is sometimes necessary due to research constraints, it is helpful to know that many males can exist in this housing set up without apparent distress.

In our study, we have attempted to establish a method for selection of potential pairs and trios of African green monkeys and described behaviors and physiological signs that can be utilized to assist in determining success of those social groups. Further research needs to be conducted to investigate which of these behaviors and the frequencies at which they are performed are the best predictors of success. Comparatively, in the heavily researched rhesus macaque, studies have shown that for both males and females, demonstrating proximity, social contact, and tandem threats (called co-enlisting in our study) on Day 1 of pairing were positive predictors of success [32]. These behaviors were also used in our study to help determine success of a social group, and further research is needed to determine if the frequency of these behaviors in African greens is similar to that seen in the rhesus. Another study in rhesus attempting to evaluate male trio formation found that out of seven trios, only one was successful but eventually also had to be separated after 51 days; it was speculated that transport stress was the ultimate reason that group failed [33]. The adult male trios in our study had many factors that could have attributed to their failures, but one that needs further research is the effect of stress on bond formation. Our trios were not transported to new locations but did have multiple episodes of sedation and study activities where separation was required and follow up studies need to be conducted to investigate to what degree the frequency of separation episodes contributes to male trio failure.

A possible solution that could assist with adult male trio formation is the addition of Diazepam during the early introductory period. In a study in adult male rhesus pairs, 16 out of 17 were successful with a dose of Diazepam, likely due to sedation and anxiolytic effects, and were successful without using the typical period of protected contact for macaques [34]. This further supports our study’s success using sedation as one of the many tools for male group formation, as well as provides an additional tool that could be used in more difficult cases.

## 5. Conclusions

African green monkeys are an invaluable alternative to macaques as an NHP research model given the critical NHP shortage, and many research facilities are now working with African greens for a variety of biomedical research objectives. There are limited publications detailing comparative behavioral differences between African greens and other NHPs in biomedical research and how best to ensure that all animals receive the appropriate and mandated social enrichment. As most of the African greens in research are of Caribbean origin, greater insight into the behavior of the African greens of St. Kitts and Nevis is a valuable addition to the biomedical field. This current analysis demonstrated that in a high-throughput contract research organization where time for in depth temperament testing for social housing is not always available, having existing data on the impact of body weight, age, and sex can be a very helpful starting point in optimizing chances of success when pair or trio housing. This information forms the basis for the behavioral, research, and veterinary teams to create compatible social groups. For females of similar ages and weights, both pair housing and trio housing was very successful. For juvenile males of similar ages and weights, pair and trio housing were very successful as well. Young adult males were most successful when paired, but trio housing can also work well, especially if enough cage space is available to allow for at least one extra cage during introductions. Adult males were most successful in pairs when at least one additional cage was provided for introductions. Adult male trios were the least successful type of grouping in the present analysis, and additional testing is needed to develop a more successful method for grouping three adult males. Assigning treatment groups based on previous social compatibility is currently the most successful strategy for forming adult male trios. Female trio housing was very successful overall and is highly recommended for any studies with treatment groups of three individuals.

The quantification of recent successes in our social housing program confirms the ongoing refinements in animal welfare at the facility which positively impacts research data emanating from studies utilizing these animals. A cooperative behavioral mindset between animal care, research, and veterinary staff is key to providing successful social enrichment at a large research organization, where all staff are invested in the success of social groups, both for enriching animal welfare and producing the highest-quality data. We hope the techniques described provide some guidelines for improved social housing at other institutions working with African green monkeys with potential relevance to other NHP species.

## Figures and Tables

**Figure 1 vetsci-11-00667-f001:**
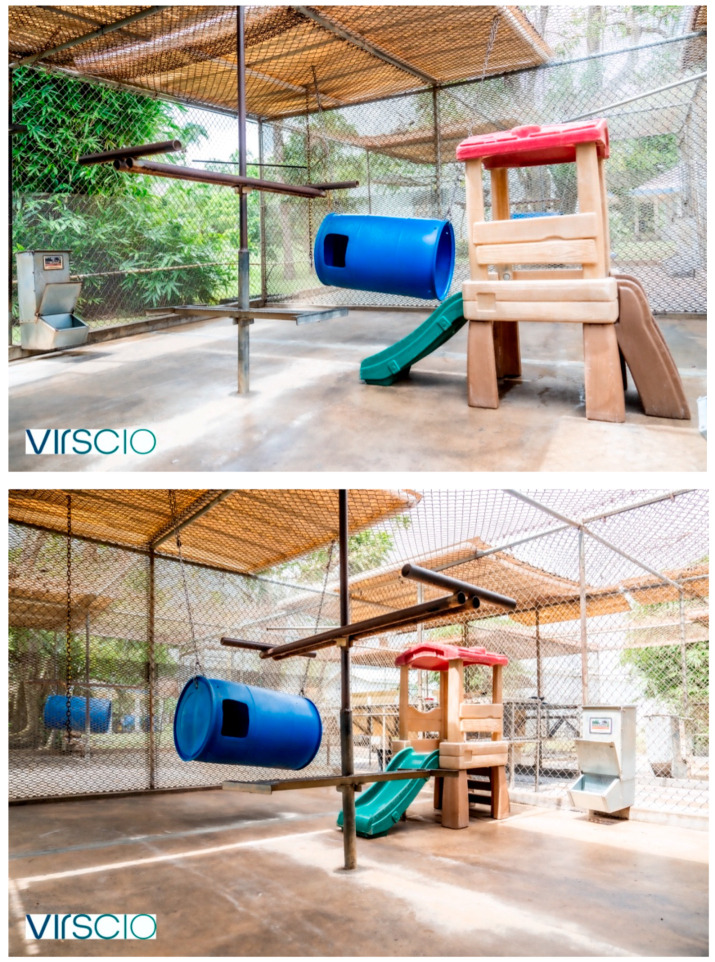
Typical social enclosure.

**Figure 2 vetsci-11-00667-f002:**
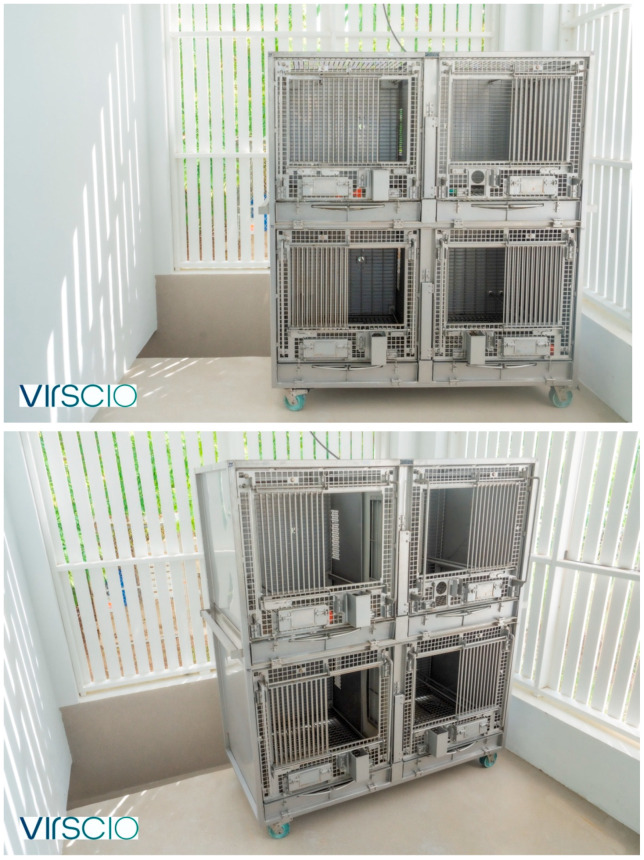
Typical quad arrangement of cages. There are a variety of options for mesh barriers between cages, with different sized holes for tactile interaction. The barriers can all be reduced to allow full contact, with a sight barrier remaining. Both central floor panels can be removed to allow animals to access and move throughout all four cages. Natural sunlight and breeze ventilation is provided by the open slatted building walls.

**Figure 3 vetsci-11-00667-f003:**
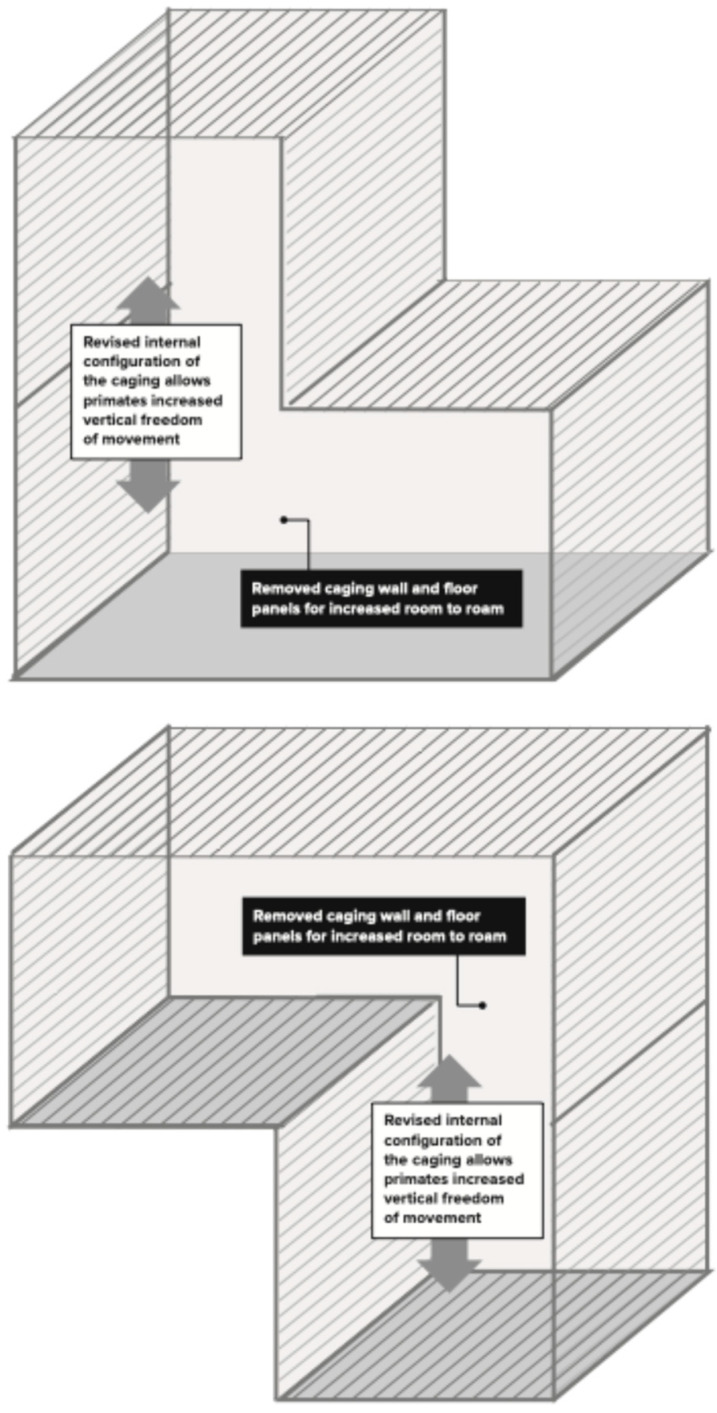
L configuration of three connected cages.

**Figure 4 vetsci-11-00667-f004:**
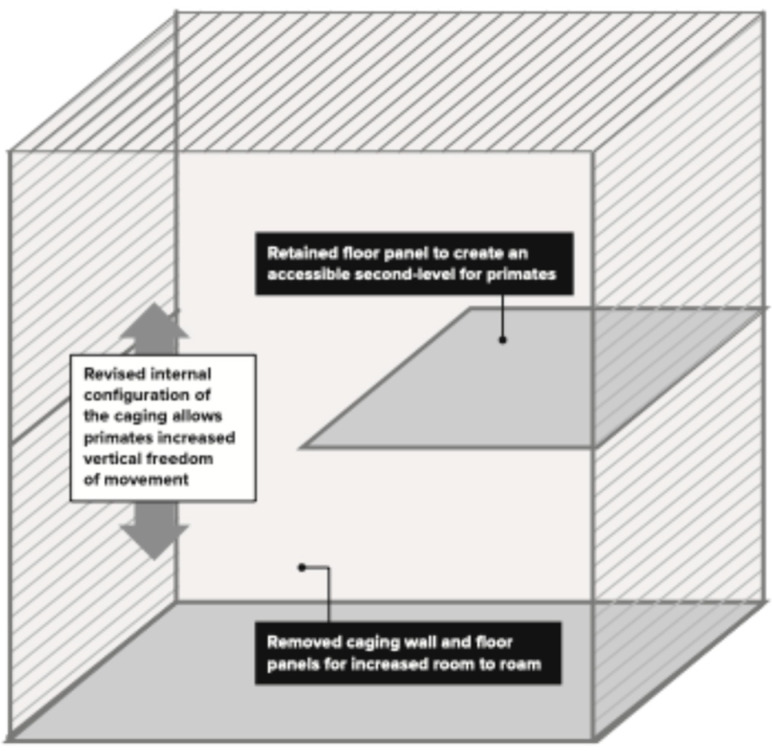
C configuration of four connected cages.

**Figure 5 vetsci-11-00667-f005:**
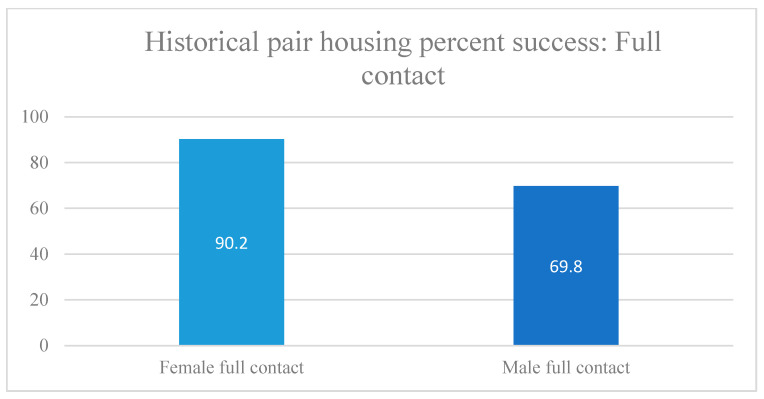
Historical pair-housing success by sex: full contact. Full contact refers to a pair of animals that are housed in two individual cages that have the center barrier removed, to allow them full access to each other (see Figure 2).

**Figure 6 vetsci-11-00667-f006:**
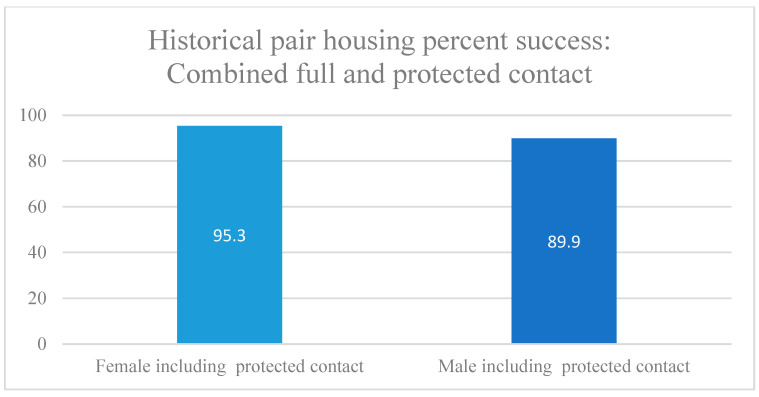
Historical pair-housing success by sex: including protected contact. Protected contact refers to a pair of animals housed next to each other in individual cages, connected by a mesh barrier, allowing the animals to have some physical contact with each other (see Figure 2).

**Figure 7 vetsci-11-00667-f007:**
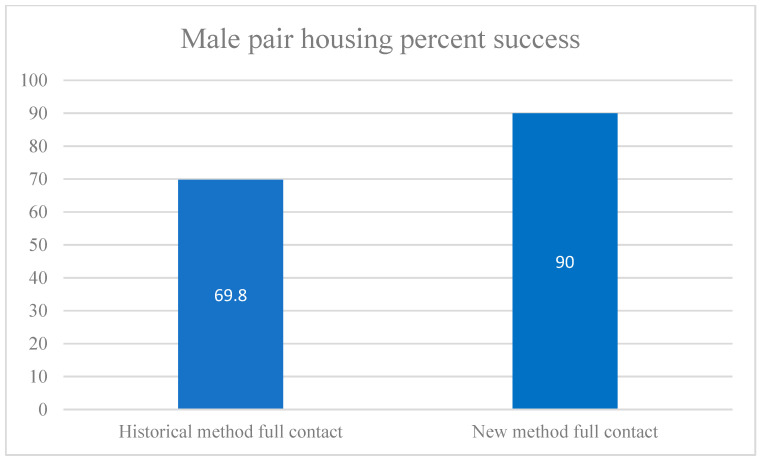
Male pair-housing success by cage size. Percentage of success of full-contact male pairs is shown above, comparing the historical method of two individual cages per pair with the new method of *n* + 1 caging (at least three cages per pair). See Figure 3 and Figure 4.

**Figure 8 vetsci-11-00667-f008:**
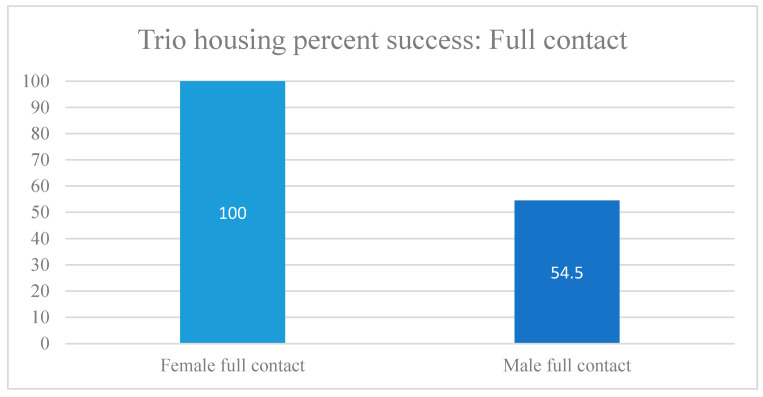
Trio-housing success by sex: full contact.

**Figure 9 vetsci-11-00667-f009:**
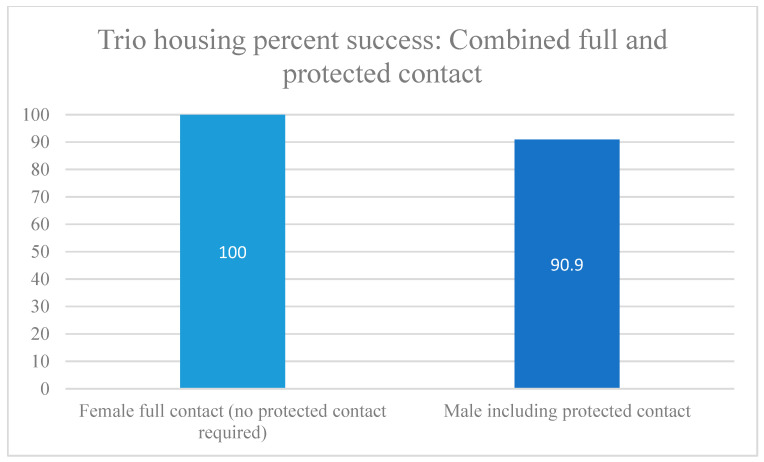
Trio-housing success by sex: including protected contact. For description of protected contact, see Section 2.3.1.

**Table 1 vetsci-11-00667-t001:** Signs of social housing compatibility. See ethogram in Appendix C for behavioral definitions.

Signs of a successful Pair/Trio include, but are not limited to:
Grooming and other affiliative behaviors
Co-eating *
Social contact/proximity
Proximity after a threat
Co-enlisting against a perceived threat
Lack of aggressive behaviors or fear behaviors
Lack of stereotypic behaviors/self-injurious behaviors **

* If the pair or trio is otherwise successful but not sharing food well, these groups can be separated just for feeding or can work with a trainer. ** In some cases, behaviors that existed prior to grouping, like flipping, may be greatly reduced by social housing but not fully eliminated if other stressful triggers cause them. In these cases, the group would not be separated unless careful observation showed that the stereotypic behaviors were initiated or increased by the presence of the partner(s).

**Table 2 vetsci-11-00667-t002:** Signs of social housing incompatibility. See ethogram in Appendix C for behavioral definitions.

Signs of social incompatibility in a group include but are not limited to:
Decreased appetite
Bloody stool/stress colitis/rectal prolapse
Hair loss/increased loose hair observed in the cage *
Visible wounds (abrasions, bruises, lacerations)
Distancing
Avoiding proximity during a threat
Prohibiting access to food or water
Prohibiting access to the full cage space
Observed threats, Contact aggression **
Stereotypical behavior ***
Self-injurious behavior
Fearful or depressed behaviors (Fear grimace, avoiding group members, lack of engagement with enrichment items)

* Can either be from an animal plucking its own hair, other animals pulling it out, or a result of an unobserved fight. Often contact aggression can happen when no humans are present in the room, so loose hair is a helpful clue for technicians. ** Minor contact aggression is the most commonly observed type performed in the presence of a human. Any significant lacerations are often caused by episodes of major contact aggression that occur overnight. No serious injuries occurred in our study, but some did warrant separation for veterinary treatment. *** If any animals were known to display stereotypical behaviors before they were placed in the social group, the technicians would observe if these behaviors increased in frequency, or if new behaviors occurred after the new grouping was formed.

**Table 3 vetsci-11-00667-t003:** Historical pair-housing success.

Total animals in the pairing program: 443 (404 wild caught, 39 colony born)	
Paired animals—separated: 32 animals	7.2% (75% *)
Paired animals—protected contact: 54 animals	12.2%
Successful pairs—full contact: 357 animals	80.5%
Total females in the pairing program: 234	
Paired females—separated: 11 females	4.7% (72.7% *)
Paired females—protected contact: 12 females	5.1%
Successful female pairs—full contact: 211 females	90.2%
Total males in the pairing program: 209	
Paired males—separated: 21 animals	10% (76% *)
Paired males—protected contact: 42 animals	20%
Successful male pairs—full contact: 146 animals	69.8%

* The percentage of separations due to contact aggression.

**Table 4 vetsci-11-00667-t004:** Summary of results for new methods of social housing. Of 25 female trios, all 25 were successful groupings, remaining together until the end of the studies (100% success full contact). Of 11 male trios, 6 were successful, remaining together until the end of the studies (54% success full contact, 90% success including protected contact). Of 20 male pairs using the new method, 18 were successful, remaining together until the end of the studies or stayed together permanently for non-terminal studies (90% success full contact).

Study	Study Type	Social Groups	Result	Cage Configuration	Comments
1	Ocular	4 female trios	Success	3 cages in an upside-down L formation	
2	Safety/Toxicology	4 female trios	Success	4 cages in a C-formation	
3	Ocular	4 female trios	Success	3 cages in an upside-down L formation	
4	Musculo-skeletal biodistribution	6 female trios	Success	3 cages in an upside-down L formation	
5	CNS	1 female trio	Success	4 cages in a C-formation	
6	Hematology	1 female trio	Success	4 cages in a C-formation	
7	Ocular	1 female trio	Success	3 cages in an upside-down L formation	
8	Ocular	1 female trio	Success	3 cages in an upside-down L formation	
9	Ocular	1 female trio	Success	3 cages in an upside-down L formation	
10	Ocular	1 female trio	Success	3 cages in an upside-down L formation	
11	Ocular	1 female trio	Success	4 cages in a C-formation	
12	Ocular	1 male trio	Failure	4 horizontal cages with sight barriers	Separated by study personnel due to multiple episodes of minor contact aggression.
13	Safety/Toxicology	4 male trios	2 Success 2 Failure	4 cages in C-formation	2 groups separated by study personnel due to an episode of major contact aggression. 1 was able to stay in a pair, with the aggressor in protected contact.
14	Ocular	3 male trios	Success	6 cages in a long C-formation	
15	Aging	2 male trios	Failure	3 cages in an upside-down L formation	Each separated due to multiple episodes of minor contact aggression.
16	Musculo-skeletal biodistribution	1 male trio	Success	4 cages in C-formation	
17	Ocular	2 male pairs	Success	3 cages in an upside-down L formation	
18	Ocular	5 male pairs	Success	3 cages in an upside-down L formation	
19	Ocular	2 male pairs	1 Success 1 Failure	3 cages in an L formation	The failure was able to stay in protected contact.
20	Small molecule	1 male pair	Success	4 cages in a C-formation	
21	Ocular	1 male pair	Failure	3 cages in an upside-down L formation	Separated due to emotional distress.
22	CNS	2 male pairs	1 Success 1 Failure	2 horizontal cages	Separated due to an episode of major contact aggression.
23	Aging	3 male pairs	Success	2 horizontal cages	
24	Ocular	2 male pairs	Success	3 cages in an upside-down L formation	
25	Ocular	2 male pairs	Success	3 cages in an L formation	

## Data Availability

The data presented in our study is available on request from the corresponding author due to privacy restrictions. All behavioral data are available; however, the details of the specific studies are restricted due to the privacy of the study sponsors.

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
