# Peer review of "Factors That Determine Successful Social Housing of African Green Monkeys (Chlorocebus sabaeus) in Same-Sex Pairs and Trios"

_vetsci, 2024, doi:10.3390/vetsci11120667_

Round 1
Reviewer 1 Report
Comments and Suggestions for Authors
The study is highly relevant and important as it aims to greatly improve non-human primate housing in testing facilities. To do so, the authors have implemented an alternative housing method where primate pairs and trios are established. Next the effectiveness of the established pairs and trios is assessed by ‘monitoring’ signs of compatibility and incompatibility which is then linked to factors like sex, weight difference and age difference.
However at this point, the study lacks clarity in the used ‘monitoring’ methods. The signs of social housing compatibility and incompatibility that are listed are still very broad and open for discussion. For this study to be relevant and usable for other facilities and the broader scientific community, a more detailed overview of the observed behaviors with their detailed definitions is crucial. Furthermore, there is a need to use more quantitative monitoring of the social housing compatibility of the primates (e.g. what is the frequency of the contact aggression observed in the initial introduction phase, and later phases?; how frequent are affiliative, stereotypical, self-injurious,… behaviors observed; does the frequency of stereotypical and self-injurious behavior increase or decrease after separation for example ).
The current study informs about the total proportion of successful pairs and trios that is based on a subjective compatibility measure, whereas there is no consideration of the behavioral repertoires or any animal-based indicators for animal welfare of the successful pairs. The authors end their manuscript with writing the following: “We hope the techniques described provide a template for improved social housing at other institutions working with African green monkeys with potential relevance to other nonhuman primate species.”. However at this point, the template needs further clarification to be usable for other facilities in improving social housing of African green monkeys.
While studies like this one can greatly contribute to the welfare of NHP in testing facilities, we should strongly advocate for science-based/ animal-based welfare monitoring to inform animal management practices. When compatibility success is solely based on ‘signs of…’ without further consideration or reporting on the frequency of relevant behaviors, also during follow-up phases, such studies risk tending towards green washing activities that are present in many other contexts.
Reviewer 2 Report
Comments and Suggestions for Authors
This manuscript discusses the important topic of African green non-human primate pairing options across gender, size, age, etc. It describes differences between macaques and African Greens and demonstrates multiple methods of pairing. It describes the most successful methods and provides new techniques to pair challenging animals. Overall, the paper is good but is very redundant and could use some additional editing to be succinct and demonstrate the data in a more typical format consistent with a scientific paper with statistics. Specific areas that could improve with additional editing:
1. There is a strange formatting change that occurs with the Materials and Methods compared with the introduction. There is no longer indentation of the paragraphs.
2. Information on intake and quarantine:
a. Too detailed information is given on the capture of the animals. Information on how the farmers are trained etc doesn’t appear to be in line with the purpose of the paper.
b. Specific information on additional screening recommended by the veterinary staff needs to be included.
c. Remove the line about each building containing a specific number of cages (lines 140-141 as it has no impact on the study.
3. The second paragraph on typical quad arrangement of cages is largely redundant and can probably be removed.
4. I’m surprised that this group is using xylazine. I don’t think that additional information needs to be included in the paper, but most non-human primate groups have switched to dexdomitor.
5. For materials and methods, I would move all the statistical analysis into one section and include sub sections. It’s difficult to read with this separated out.
6. Presentation of results:
a. Table 3 could use more details and relate them to the signs of social incompatibility seen in table 1b. If not relatable, I’m not sure this information needs to be in a chart or could percentages be given?
b. There doesn’t seem to be any presentation on how historical methods are successful vs the new methods.
c. The graphs are missing information about statistics. While the stats are in the results, it is much easier to read if that information is visual in the chart and described with the Figure as well.
d. Tables 4, 5, and 6 could probably be combined and I would still like to see how these are related to table 1b. Additionally, it’s not clear if they are trying to demonstrate that the type of study influences the outcome?
e. Statistics between males and female group success should be determined.
Reviewer 3 Report
Comments and Suggestions for Authors
Dear authors,
Thank you for submitting this interesting paper. Most research concerning social housing and related challenges are published in macaques. African Greens are often neglected as research animal. Social housing improves animal welfare. Hopefully, the result of this publication will be that other working with African greens will also go for social housing.
I red the manuscript with great pleasure and can write that by publishing useful work. The level of English is perfect and although it are many pages, it reads smoothly.
Off course I have remarks:
- I would love to see in the discussion a paragraph to compare your results with more publications in macaques then only Truelove et al.,2017 like:
o MacAllister RP, Heagerty A, Coleman K. Behavioral predictors of pairing success in rhesus macaques (Macaca mulatta). Am J Primatol. 2020 Jan;82(1):e23081. doi: 10.1002/ajp.23081.
o Logan LE, Sayers K. Pairing Laboratory-Housed Adult Male Rhesus Macaques (Macaca mulatta): Success Rates in Relation to Behavioral Response and Duration of Visual Contact. Appl Anim Behav Sci. 2024 Aug;277:106340. doi: 10.1016/j.applanim.2024.106340.
o Ruhde AA, Baker KC, Russell-Lodrigue KE, Blanchard JL, Bohm RP. Trio housing of adult male rhesus macaques (Macaca mulatta): Methodology and outcome predictors. J Med Primatol. 2020 Aug;49(4):188-201. doi: 10.1111/jmp.12469.
o Kezar SM, Baker KC, Russell-Lodrigue KE, Bohm RP. Single-dose Diazepam Administration Improves Pairing Success of Unfamiliar Adult Male Rhesus Macaques (Macaca mulatta). J Am Assoc Lab Anim Sci. 2022 Mar 1;61(2):173-180. doi: 10.30802/AALAS-JAALAS-21-000059
o This to show that your results could be applicated in macaques too.
- Or maybe I want to write that in the discussion, no references are used! Normally the obtained results are compared and explained with other references but in your conclusion, no references are used. Please adapt
Further some minor remarks:
- In the abstract the latin name behind African green in line 23 should be added
- I suggest, as 10 key words are allowed, to add the words environment, enrichment, welfare, and refinement to the keywords
- I would suggest to rewrite line 35/36 to “The current shortage of research non-human primates (NHPs) has elevated interest in African green monkeys (chlorocebus sabaeus) [1].
- Lines 232, 519, and 568 NHP should be used
- In line 45 you write ‘to the facility’ without explaining what the facility is. Maybe you can keep it general here and create something like ‘..for African green monkeys housed in medical research facilities’.
- Line 53: ; should be a . and However with a capital
- I suggest to delete line 69-70 as it seems useless
- The flow of the introduction seems African greens, wildcaught, behavior, social paring and hypothesis. Therefore, the paragraph=lines 72 to 85 can be totally deleted. This information is not needed.
- Maybe I missed it but in the text the reference to Figure 1a and 1b is missing
- Line 100: humanely caught should be old fashioned wild-caught as that is the general used term.
- In the quarantaine: I assume ID is also done with tattoo and chip? Add.
- In the quarantaine: I assume single housed? Please add. And add how long-days, your quarantaine lasts and the exact number of TB testing.
- Around line 140: add refer to Figure 2
- Line 174: delete light as 8mg/kg ketamine is not light
- Line 176: wake upn together is in veterinary language recover, line 177 idem, fully awake is fully recovered
- Line 188: move info name of your database from line 259 to here as this is the first mentioning.
- Table 1b. In my opinion: Weight loss is not a good sign to score as it takes a long time to loose enough weight to be observed and body weight is hard to judge from outside the cage so without palpation. Fearfull behavior: lack of engagement with enrichment behavior says nothing about fear as it could just be lack of interest to the enrichment, boring enrichment.
- Line 232-235: far too long sentence. ; could be a dot and start the new sentence with However
- Line 237: the animals are monitored ….
- Line 278: what is the additional value of writing ( n+1)? As in the sentence it is explained which cages were provided and it says sometimes 4 cages so n+2 were provided.
- Line 279-280: if figure 4 is earlier mentioned in the text than figure 3 the figure 4 should be named figure 3 and number 3 becomes 4.
- Line 304: three instead 3
- Liner 310: rephrase, maybe use n=25 and n=11
- Line 313: delete light
- Line 317: the Table could be moved upwards as it is now showed in line 418 while its mentioned in line 317 already.
- Line 279-280: if figure 4 is earlier mentioned in the text than figure 3 the figure 4 should be named figure 3 and number 3 becomes 4.
- Line 331: delete light
- Line 395: replace ; by a .
- Line 400-401: I would suggest to make two sentences instead of one long.
- Line 410-412: too long sentence
- Line 412: 5 should be five
- Line 441: double spacing before [16
- Line 486: delete the line ‘the final housing……trio housing.’. And Trio housing was established in 2022 please add where/which location/in which institute it was established.
- In the manuscript isosexual is used. Mostly ‘animals from the same sex’ or ‘same-sex pairs’ or ‘same-sex couples’ is used. Please adapt throughout the manuscript.
- Line 492: , after difference
- Line 507-510: too long sentence, please adapt.
- Line 516-518: too long sentence, please adapt.
- General: a lot ‘found’ and ‘findings’ is used while one can also choose ‘observed, assessed, measured,….observations,…
- Line 545-548: too long sentence, please adapt
To avoid advertising or commercials, the authors need to consider:
In line 99: “Virscio is a contract research organization…….” could be rephrased to ‘the animals were housed at a contract research organization…”
Figure 1 says “Figure 1. Typical social enclosure at Virscio“ could be deleted as the text “There are currently about 700 animals housed in large outdoor social enclosures, which range in size from 10 ft x 16 ft x 8 101
ft to 20 ft x 16 ft x 8 ft, with many being 10 ft x 16 ft x 10 ft. These enclosures are made of chain link fencing, with sealed concrete 102
flooring and shorter raised concrete walls as wind breakers. In addition to surrounding trees, each enclosure has an opaque fiberglass 103
roof on half of the ceiling to allow shade and protection from the rain as well as direct sunlight from the uncovered portion of the 104
roof. There are multiple stainless steel bars for perching, as well as hollow hanging plastic barrels with a window, to allow for both 105
a perch and sight barrier. Other forms of enrichment, such as hanging ropes, hammocks, plastic perches, plastic play structures and 106
various manipulanda are also provided to promote natural behaviors.” This text and Figure 1 are not really needed as the paper goes about pair and trio housing and not how they were housed in the time before that period. To show Figure 1 and the text= advertising
Line 120-121: “…….supports these local farmers, a number of whom are training and employed as trappers, improving the livelihood of the farming community while benefitting both biomedical research and local ecosystem health and management” is an advertisement line which is not related to the pair housing study.
Line 497/498: “As is the nature of many of the research studies conducted at Virscio” could be contract research organization
After those revisions I don’t see any problems anymore.

Round 2
Reviewer 2 Report
Comments and Suggestions for Authors
The changes made to the formatting and organization of this paper have significantly improved the readers ability to read this paper.
Author Response
Thank you so much for the comments provided! I greatly appreciated the thorough comments and recommendations and I am glad that the paper is easier to read now.
Reviewer 3 Report
Comments and Suggestions for Authors
Dear authors,
Thank you for implementing most of my suggestions. However, some minor comments are left:
- The title: ‘assessment of determining factors’ or ‘factors that determine’
- Line 11-13: to long sentence, try to use sentences of max 15-20words long
- Abbreviation of NHP in simple summary can be deleted
- Line 13: like all NHPs can be deleted
- Line 14: behavioral enrichment is a husbandry concept so incorrect use. Maybe replace it for welfare
- Line 14-17: to long sentence which is incorrect with the verbs. Please create sentence of 15-20 words.
- Line 15: environment? Do you man cage size? As in line 132 you write you want a novel enclosure
- Line 21: some can be deleted
- Line 21-24: to long sentence
- 3 should be three
- Line 25: environment. Does that mean: cage size? Cage enrichment? I assume you don’t mean the outside of the cage so please rephrase
- Line 44: medical? Biomedical? Or just delete medical and keep it on research facilities?
- Line 46: delete natural
- Line 47: enrichments provided??? Maybe better should be available? Possible? Its more decent not to call it enrichment but a standard available issue.
- Line 47: delete like all other NHPs
- Line 50: ‘effective’ can be deleted
- Line 64-67: too long sentence
- Line 69: at this facility should be ‘Studies are designed to have….’
- Line 69-83: cage size as factor is not mentioned!
- Subheader 2.2 is missing
- Line 133-137 you sedate the animals because they need to be relocated and in line 136 you write they recover in an enclosure which is novel to the group members. Confusing. Secondly, whom are the group members? Are that the new animals they are introduced too? Unclear.
- Line 138 ‘succes of the group’ is not a objective definition. What does success of the group mean? They tolerate each other? Best friends? How scored? Just in line 148 this definition comes. Please adapt.
- Table 1A and legend: sharing food is a weird scoring name as they are not humans. What does ‘sharing food’ mean? One monkey grabs an apple, takes a bit and offers the rest to the other monkey? Please adapt
- Table 1b: preventing others from getting, maybe think about ‘don’t provide others access to’ or ‘don’t allow others to take food or drinks’ as ‘preventing’ is not the correct word
- Line 244: delete lighty as the mentioned dose is the same as used earlier
- Line 236-253: I think when you write 3 animals, 2 bottom cages etc this should be three, two etc
- Line 324: delete . before (100%
- Line 331: rephrase, start the sentence with: 18 out of 20 male pairs using the new method were successful…
- Line 401: delete healthy
- Line 427: saw should be observed
- Line 428: believe could be assumed
- Line 446: it was found. Maybe replace found by a more scientific word like observed, assessed, measured, determined…..idem line 453
- Line 470: keep one line in using references so rephrase the line without using the name Jorgenson in the text but just use the number 27 at the end of the sentence.
- Line 475-478: too long sentence, please adapt.
- Line 504: add space before [34]
- Line 531: earlier you wrote ‘in this study’ and here in our social housing program. Our sounds better then this so maybe change all in this study to in our study like line 483 and 386.
- Details: references are not in the correct format I,.e. italic journal name, bolt year and italic journal issue. Some journals are written fully while others are abbreviated.
Thats it. Just minor issues.
Round 3
Reviewer 3 Report
Comments and Suggestions for Authors
Dear authors,
the revised version looks nice. One could only think abouyt adding in the title: Assessing the factors that determine successful.......
Well done!